# Mechanical cleaning of graphene using in situ electron microscopy

Peter Schweizer[1,4], Christian Dolle [1,4], Daniela Dasler[2], Gonzalo Abellán[2,3], Frank Hauke[2], Andreas Hirsch [2] & Erdmann Spiecker [1✉]

Avoiding and removing surface contamination is a crucial task when handling specimens in any scientific experiment. This is especially true for two-dimensional materials such as graphene, which are extraordinarily affected by contamination due to their large surface area. While many efforts have been made to reduce and remove contamination from such surfaces, the issue is far from resolved. Here we report on an in situ mechanical cleaning method that enables the site-specific removal of contamination from both sides of two dimensional membranes down to atomic-scale cleanliness. Further, mechanisms of re-contamination are discussed, finding surface-diffusion to be the major factor for contamination in electron microscopy. Finally the targeted, electron-beam assisted synthesis of a nanocrystalline graphene layer by supplying a precursor molecule to cleaned areas is demonstrated.

[1] Institute of Micro- and Nanostructure Research (IMN) and Center for Nanoanalysis and Electron Microscopy (CENEM), FAU Erlangen-Nürnberg, Cauerstr. 3, 91058 Erlangen, Germany. [2] Department of Chemistry and Pharmacy and Joint Institute of Advanced Materials and Processes (ZMP), Chair of Organic Chemistry II, FAU Erlangen-Nürnberg, Nikolaus-Fiebiger-Str. 10, 91058 Erlangen, Germany. [3] Instituto de Ciencia Molecular (ICMol), Universidad de Valencia, Carrer del Catedrátic José Beltrán Martinez, 2, 46980 Paterna, Valencia, Spain. [4] These authors contributed equally: Peter Schweizer, Christian Dolle. ✉email: Erdmann.Spiecker@fau.de

Contamination can be a pitfall to any scientific experiment, technological process, or even forensic investigation. Therefore, cleanliness and cleaning procedures are continuously improved and discussed in most fields of science and technology. Examples for this range from wafer cleaning in semiconductor manufacturing[1] to avoiding DNA cross-contamination in forensic science[2,3] and sterilization of spacecraft in interplanetary travel[4] to avoid spreading terrestrial life to foreign worlds. In Materials science, cleanliness has become a forefront issue with the advent of 2D materials[5], for which what is considered to be clean has to be re-evaluated. In a single-atom-thick membrane, graphene being the most prominent example[6], even a single ad-atom can make a huge impact by, for example, initiating a rapid decomposition process[7]. In addition, the characterization of such thin sheets or the targeted synthesis of novel structures on their surface becomes challenging with even a few contaminant molecules. Contamination is invariably introduced during synthesis, processing, and storage, and is therefore always a concern. While a plethora of cleaning methods have been developed for graphene in recent years, so far none was able to demonstrate a site-specific, fully cleaned surface down to the atomic scale. Techniques such as heating[8], plasma treatment[9], laser cleaning[10], chemical activation[11], and current-driven cleaning[12] have been tried out and tested. In addition, first steps toward mechanical cleaning have been made using AFM[13–15], but without achieving atomically clean membranes. While some methods (especially aging at elevated temperatures in ultra-high vacuum) have produced reasonable results, there is no control of the size and location of clean areas. In addition, many samples, such as chemically functionalized graphene, and experiments are sensitive to the energy introduced by the cleaning method.

In this work, we introduce a site-specific mechanical cleaning approach combined with in situ electron microscopy. The method can clean both sides of suspended two-dimensional membranes achieving atomically clean areas of several µm² within minutes. The method does not rely on introducing thermal energy or reactive species. Furthermore, we discuss the origins of recontamination, and the fundamental limit of cleanliness that can be achieved. We find that in electron microscopy surface diffusion is the major factor for recontamination. Finally, we use the cleaning technique in combination with the obtained knowledge about recontamination mechanisms to synthesize nanocrystalline monolayer graphene in situ in transmission electron microscopy (TEM).

## Results

**In situ mechanical cleaning setup**. Figure 1 shows the basic premise of the cleaning method as well as the technical implementation in both scanning electron microscopy (SEM) and TEM. Mechanical cleaning works in a similar way to a broom on the macroscopic scale. By moving a broom over a surface, loosely bound dust and dirt are removed. This principle can be transferred to the nanoscale (see Fig. 1 for an artistic representation) where the broom becomes a single fine-metal tip controlled by a piezo-driven nanopositioning system. This tip effectively separates adsorbed contamination from the graphene surface (Fig. 1b) in an area around it. A certain amount of normal force is required to ensure a direct contact between the tip and graphene. The method works the same way on the both sides of a suspended membrane. The underlying physical principle behind the method is the stark contrast in bond strength between the strong intralayer bonds of the atoms in the graphene sheet compared with the much weaker van der Waals type bonds between contaminating molecules and graphene. Those bonds are essentially unzipped by

the cleaning process (Fig. 1c). In this schematic PMMA residuals are shown since the polymer is commonly used in the procedure of transfer of graphene (and other 2D materials) and is never entirely removed by conventional methods[8]. Nevertheless, the method also works for other types of contaminants.

The technical implementation was done in both SEM and TEM to demonstrate the viability in different environments and make use of the distinct advantages of each instrument. Due to the larger accessible volume in the vacuum chamber of SEM, two independent micromanipulators were used (Fig. 1d), one dedicated for each surface side. Each manipulator has three independent degrees of freedom, and is outfitted with a tungsten metal tip (tip-radius 50–150 nm, the tips were taken "as-is" without further treatments). For the TEM implementation, the setup needed to be reduced in complexity because of the limited available space. Using a nanofactory STM holder, a single three-axis micro-manipulator is available (Fig. 1e). To reach both sides of a suspended membrane, the manipulator is outfitted with a crescent-shaped metal double tip, which is made from a gold wire via electrochemical etching followed by focused ion beam (FIB) milling. The setup in the holder as well as a photograph of a representative tip is shown in Fig. 1f. The sample size is limited in this setup to a half-grid with a diameter of 3 mm.

**Cleaning process in SEM and TEM**. A freshly prepared, suspended graphene membrane is covered in organic residuals even after careful washing and pre-cleaning. The residual contamination is shown in Fig. 2a, for SEM, and Fig. 3a for TEM imaging. Due to the surface-sensitivity of secondary electrons, the contamination is clearly visible in SEM, whereas in TEM only a faint contrast is obtained due to the weak scattering amplitude of carbon for high electron energies. One of the tips used for cleaning is approached to the surface of the suspended membrane until it is in direct contact. The approach is performed using visual guidance only. The depth of field is used for rough positioning; moving into contact is then performed by slowly approaching the tip further until a subtle movement of the membrane is visible. By moving the tip in a sweeping motion (Figs. 2 and 3b), the contamination is removed. This process is repeated with the second tip on the opposite side (Figs. 2 and 3c) until the suspended membrane is fully cleaned (Figs. 2 and 3d). Cleaning works irrespective of layer number with bi- and trilayer graphene being cleaned in the same way. The contamination tends to stick to the cleaning tip where it can even aid in the further cleaning process by increasing the effective contact area. The effect of increased area by accumulated contamination can easily be seen in both examples. Larger pieces and more rigid contamination are more likely pushed to the sides as a whole and stays on the samples (see below). See Supplementary Movies 1a, b, and 2 for the whole-cleaning process. In the example, the process took ~6 min in the SEM and ~11 min in TEM. The main limiting factor in SEM was the refresh rate which was kept to ~0.5 per second to obtain high-quality images. With faster scan speeds, repeated cleaning times of just ~1 min are achievable. In TEM, the sequential approach with the double tip proved to be the time-limiting factor. By tilting the sample by ~80° in SEM, a perspective view of the cleaning process can be gained (Fig. 2e–h; Supplementary Movie 1a). This view shows little deflection of the graphene membrane as a whole during cleaning, while localized topography in the form of wrinkles can form around the tip. To show that the cleaning process does not introduce additional defects into the graphene membranes, Raman spectroscopy was performed after cleaning. Similar spectra are obtained for both the as-prepared as well as the cleaned membranes (Fig. 2i). In both cases, the G-band is clearly resolved, with the D-band only

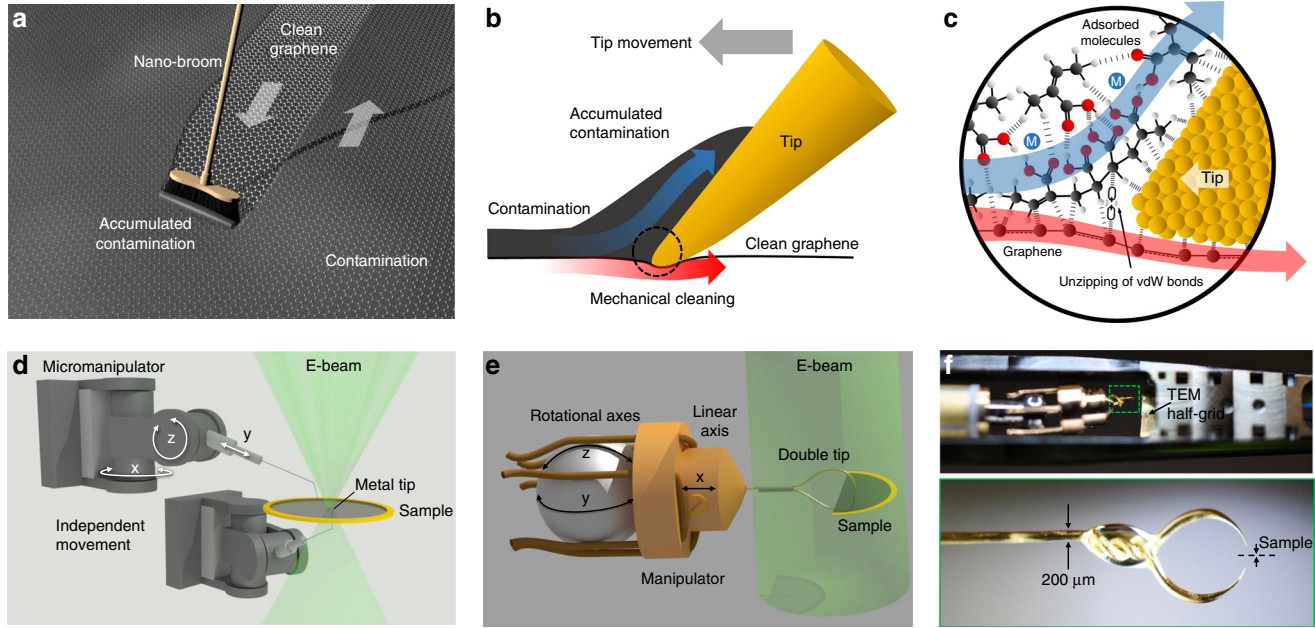

**Fig. 1 General concept and technical implementation of mechanical cleaning. a** Artistic representation of the cleaning approach: the mechanism of cleaning with a broom is transferred to the nanoscale. **b** The "broom" becomes a small metal tip that separates adsorbed contamination from the graphene substrate. **c** The tip unzips weak van der Waals type bonds between contaminants and the substrate while the strong intralayer bonds of graphene stay intact. **d** Setup in the SEM: two independent three-axis micromanipulators are used, one dedicated for each side of a suspended membrane. In TEM, only a single manipulator is available (**e**) due to size constraints. This manipulator is therefore outfitted with a crescent-shaped double tip to reach both sides of suspended membranes. **f** Sample and manipulator in the TEM holder (top) and a photograph of a representative double tip (bottom).

slightly showing up in the as-prepared state. This could be attributed to a fold in the membrane in the selected window. The slight increase in signal between $1000\,\mathrm{cm}^{-1}$ and $1600\,\mathrm{cm}^{-1}$ can be attributed to unavoidable background originating from the support film (see Supplementary Fig. 1 for a reference spectrum of the support film and details on the measurement). The 2D band does not show a significant shift or broadening.

In situ force measurements were performed during cleaning to determine thresholds for both the lateral and normal force required for removing contamination. For the measurements, a sample is placed on a spring table that deflects under load. This small deflection can be tracked using automated template matching and used to calculate the force in one spatial direction. In order to obtain both lateral and normal forces, two different measurement setups were used corresponding to a birds-eye view and the aforementioned perspective view (see Supplementary Fig. 2 for more detail on the setup and measurements). Figure 2j shows a lateral force curve, acquired during cleaning. A consistent value of ~76 nN was measured during cleaning as shown in Fig. 2g. The sign change of the force signal is attributed to the direction of movement which follows a sweeping pattern. The lateral cleaning force $F_c$ consists of two contributions given by:

$$F_c = F_f + F_a$$

with the friction force $F_f$ and the adhesive force $F_a$. The friction force is assumed to be well below 1 nN (as it was found for tips and small carbon patches on graphite using friction force microscopy[16,17]) leaving the adhesion between graphene and the contamination as the major contribution to the cleaning force. Neglecting the contribution from friction, the fundamental adhesion $E_a$ energy can be estimated according to:

$$E_a = F_a \cdot \frac{\mathrm{d}x}{\mathrm{d}A} = 0.36 \pm 0.05\,\mathrm{J\,m^{-2}}$$

with the change in tip position $\mathrm{d}x$ and the change in free surface

area $\mathrm{d}A$. The resulting value of $0.36\,\mathrm{J\,m^{-2}}$ is close to experimental results on the cleavage energy of graphite ($0.37\,\mathrm{J\,m^{-2}}$[18]). Experiments on the desorption energy of organic molecules also found values in the range of $0.35\,\mathrm{J\,m^{-2}}$[19], which agrees well to our findings. The minimal normal force required for cleaning was determined to be ~50 nN for as-prepared membranes. This value is usually far exceeded during cleaning (up to 1.6 µN) to ensure a good contact throughout the process. However, in all measured cases, the normal force stayed well below the threshold for membrane rupture (around 7.5 µN for the used tips, see Supplementary Fig. 3 for a measurement of the strength of graphene membranes). The force required for cleaning is influenced by the structure of the contamination layer. If the contamination is strongly interconnected and can only be removed in bigger chunks at the same time (see Supplementary Fig. 4), the lateral force needed for cleaning is also increased. We find a general relationship of $6.7\,\mathrm{µN\,µm^{-2}}$ (MPa) for the removal of arbitrarily sized contamination. Interestingly, this value is close to the value found for the interlayer shear strength of graphite[20].

We can therefore conclude that the in situ mechanical cleaning method can selectively remove adsorbed contamination resulting in atomically clean membranes. The forces required for cleaning are usually well below the yield point of pristine graphene. In the presence of defects, the yield strength of graphene can be considerably lowered[21,22], which may lead to the premature rupture of membranes in some cases. Another effect is the strengthening of the contamination layer by increased thickness and chemical cross-linking under e-beam irradiation[23]. Nevertheless, overall success rates of >80% were still achievable on freshly prepared membranes.

To confirm the level of cleanliness, high-resolution transmission electron microscopy (HRTEM) was performed directly after cleaning in the TEM. The images (see Fig. 3e) reveal a pure graphene lattice without any residual contamination or mechanical damage as already hinted at in Raman spectroscopy. The only

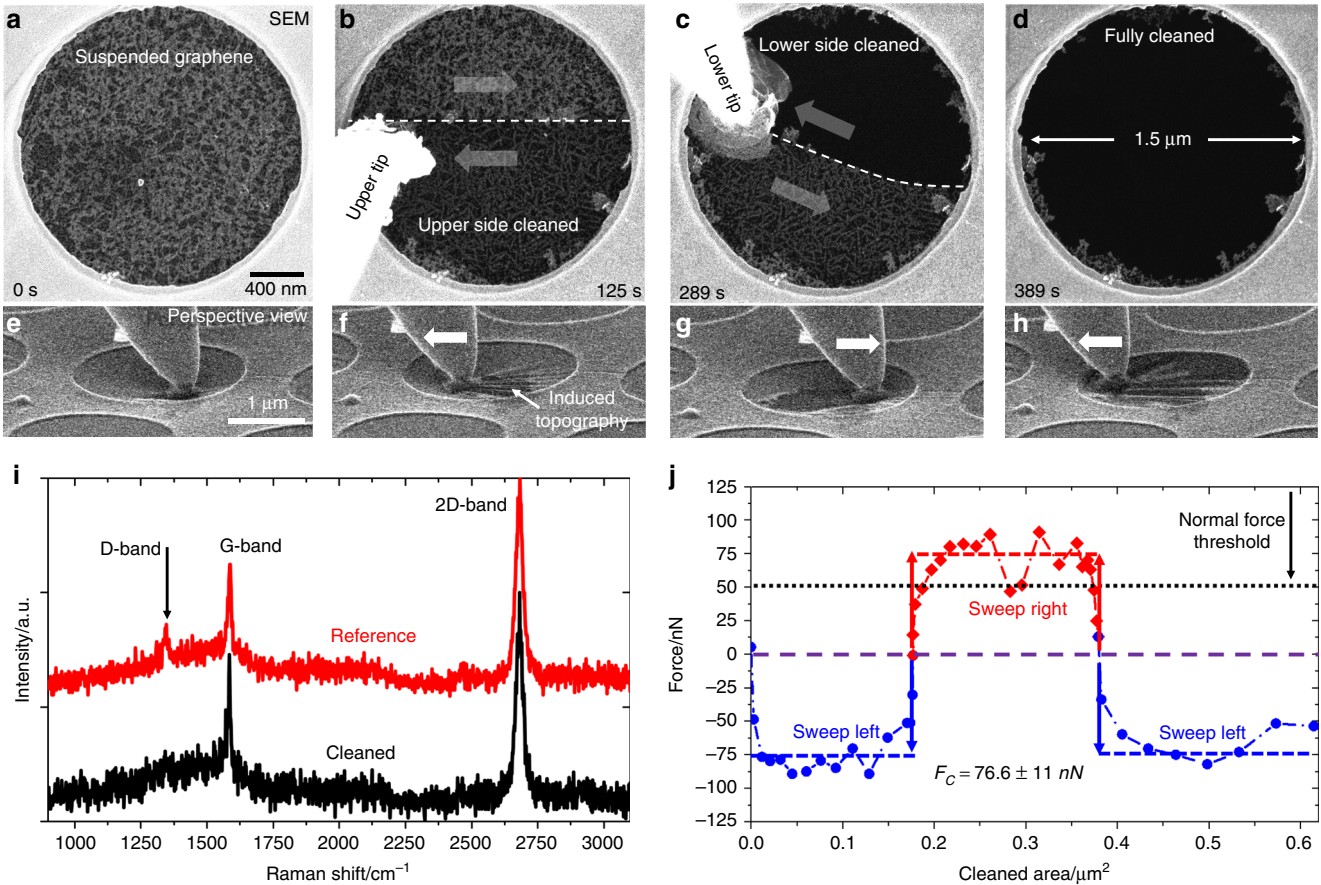

**Fig. 2 In situ mechanical cleaning procedure in SEM. a** A suspended graphene membrane with residual contamination as seen in SEM. By lateral movement of a fine tip, the contamination can be removed first from one side (**b**) and subsequently from the other side (**c**). After the process, a fully cleaned membrane is obtained (**d**). **e–h** Perspective view of the cleaning process (one side only) showing the introduction of topographic wrinkles in the membrane. The full cleaning process can be viewed in Supplementary Movies 1a, b. Raman spectroscopy (**i**) shows no increase in D-band intensity due to the cleaning process and no shift of the band positions. **j** The lateral force required for cleaning was found to be in the range of 76 nN with a minimal normal force of ~50 nN.

features found on the membrane were intrinsic defects such as divacancies (Fig. 3f). To learn more about the origin of contamination on the membrane, electron energy loss spectroscopy was performed on as-prepared and mechanically cleaned membranes (cleaned in SEM and transferred to TEM to include airborne contamination). In the as-prepared spectrum, a clear oxygen signal was found which vanishes after cleaning (see Fig. 3g). Since graphene was transferred to the TEM grids using PMMA, it stands to reason that residuals from this polymer are the origin of the oxygen signal. PMMA has also been made out as a major contamination source in other studies[8,24]. A second effect of the cleaning procedure is a shift of the plasmon loss peak from ~22 eV to ~16 eV (Fig. 3g, inset). While the former value fits well to the plasmon signature of PMMA[25], the latter is more indicative of pristine monolayer graphene[26]. Even though the cleaned sample was exposed to ambient conditions and is therefore repopulated with contamination originating from air, the carbon K-edge fine structure is much better resolved due to the reduced thickness of the contamination layer. Airborne contamination is mostly comprised small hydrocarbon molecules that adsorb on the graphene surface[27].

The immediate benefit of clean membranes in the context of electron microscopy lies in long-term stability (see Supplementary Fig. 5 for a comparison of the dissolution rates with and without cleaning) and the ability to use low energies. Low energy imaging enabled by in situ cleaning has already been exploited in

prior work for the direct manipulation of defects in bilayer graphene[28,29]. While graphene serves as the benchmark to showcase the method, it can be readily applied to other 2D materials like $MoS_2$ (see Supplementary Fig. 6).

**Recontamination of atomically clean membranes**. When a surface is cleaned, the real challenge is keeping it clean. To learn more about recontamination mechanisms, both ex situ and in situ experiments on cleaned membranes were conducted. To see the effect of airborne contaminants, which can have a profound effect on the properties of graphene[27], a mechanically cleaned membrane (shown in Fig. 4a) was exposed to ambient conditions for 5 min before it was put into the high-vacuum environment of a TEM. Looking at the once-cleaned membrane, a complete layer of recontamination is found. The layer is discontinuous with small patches of clean areas remaining. Compared to as-prepared specimen, there is little improvement in the overall cleanliness when subjected to ambient conditions. Interestingly, in the case of inserting the specimen into the vacuum chamber of the SEM, the contamination layer was continuous without clean patches in between (see Supplementary Fig. 7 for images). This can be attributed to the difference in evacuation speed and vacuum level ($10^{-5}$ mbar for SEM and $10^{-7}$ mbar for TEM). It seems that a sufficiently high vacuum alone can break up contamination layers on the membranes, but not remove it entirely. Exposure to

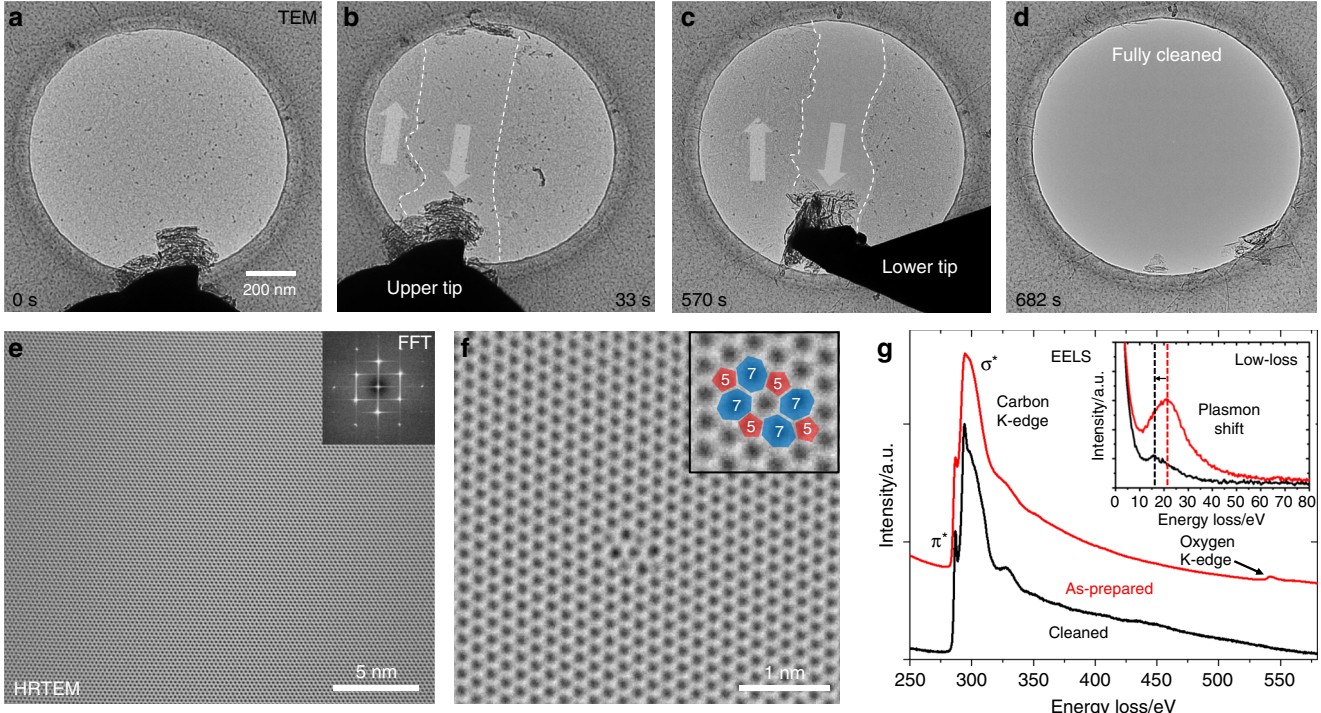

**Fig. 3 In situ mechanical cleaning approach in the TEM. a** TEM image of a graphene membrane with residual contamination. **b, c** Consecutive cleaning from both sides results in an atomically clean membrane (**d**). HRTEM (**e**) confirms the level of cleanliness and pristine state of the graphene. The FFT (inset) confirms to the graphene structure. Only intrinsic defects such as divacancies (**f**) are found on the membrane with no additional defects being introduced by the cleaning method. **g** EELS spectra of as-prepared and (SEM) cleaned graphene membranes. The oxygen signal from the PMMA residuals disappears after cleaning. Moreover, the plasmon loss peak decreases in intensity and shifts to lower energies.

ambient conditions does inevitably repopulate a clean area with contamination.

A different issue is in situ recontamination while the sample remains in the vacuum environment. In electron microscopy, the growth of contamination induced by the electron beam has long been a well-known phenomenon[30]. With atomically clean membranes, it is now feasible to study the mechanisms of this process in detail. Exposing such a clean membrane (see Fig. 4b) for a certain period of time (~6 min) to a moderate electron beam (20 kV, 69.1 µA mm$^{-2}$) results in the growth of contamination seeds. Those seeds are only few nm in size and have an amorphous appearance in HRTEM (Fig. 4b, bottom). A preferential formation of seeds at defects, such as dislocations in bilayer graphene is observed (see Supplementary Fig. 7). Over time, the seeds start to grow while new seeds can still form. This leads to an accelerating growth of the recontamination layer which finally results in a coalescence of contamination. Rotating an in situ recontamination data cube results in space-time slices, revealing the growth of seeds over time as shown in Fig. 4c. The time runs from left to right with a spatial coordinate running from top to bottom. The original data are found in Supplementary Movie 3. In this example, initially few contamination seeds are present which grow linearly over time while others appear concurrently. While the seeds are still small, they may fluctuate in position over the membrane (example marked with arrows) until they are immobilized. By calculating the covered area fraction for each frame of the in situ movie, a contamination plot over time is obtained (Fig. 4d). The contamination rate stays low during the beginning with only few seeds being present. After ~60 s, a linear growth regime is reached with seeds growing in size. A change in scanning speed induces the formation of additional seeds, leading to an accelerating growth of the covered area. Both in situ and ex situ recontamination can be removed repeatedly by

mechanical cleaning (see Supplementary Movie 4). The growth of a contamination layer necessitates the supply of precursor molecules that are fixated by the electron beam. To test their origin, scan-time experiments were conducted: a cleaned area was illuminated with a high electron dose (1.4 mA mm$^{-2}$) with a very low scanning speed (30 s for a single pass over the area) and compared with a second area which was illuminated with a very high scanning speed (~0.7 ms for a single pass, 50 ns dwell time, same dose rate). The accumulating contamination was tracked over time using the HAADF signal in STEM mode. For the low scanning speed, a uniform increase in the contamination layer is observed. On the other hand, the fast scanning speed results in a thick layer at the sides of the pattern with a much lower recontamination rate in the center of the pattern (see Fig. 4e for x-t slices). The data cubes have also been rendered in 3D for illustrative purposes (Fig. 4f), highlighting the difference in recontamination rate. Based on these results, the processes happening during in situ recontamination can be described as shown in Fig. 4g. Recontamination is governed by the immobilization of mobile organic species. This occurs due to electron-beam-induced cross-linking of two smaller molecules to a bigger one. For the origin of those molecules, there are two options: either gas-phase or surface diffusion. If gas-phase diffusion were the prevalent source, recontamination would be uniform independent of scan speed due to the uniform supply of species. However, our experiments show that this is not the case with, instead, surface diffusion being the dominant mechanism which has also been previously suggested[31]. Molecules moving by the process of surface diffusion can only move into the illuminated areas from the sides of the scanning pattern. If the scanning speed is high enough, they are "caught" by the beam and (given a high enough dose) fixated before moving further inside the pattern. With different-sized species and by random chance,

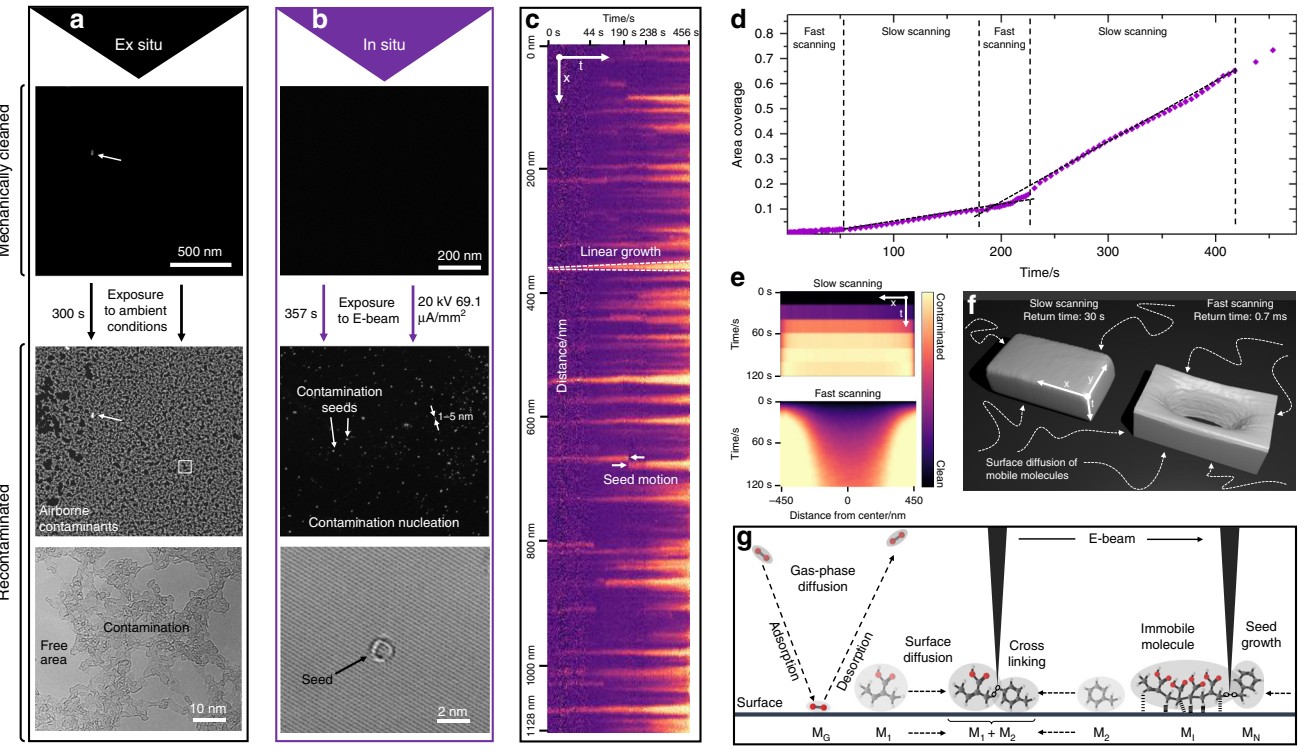

**Fig. 4 Recontamination mechanisms. a** Subjecting a cleaned area to ambient conditions for five minutes results in a coverage with amorphous contamination. Small patches are still clean enough for HRTEM investigations. **b** During illumination of a cleaned area with the electron-beam contamination seeds start to form. The seeds are few nm in size and of amorphous appearance. **c** x-t slice of a recontamination experiment (see Supplementary Movie 3 for the whole process). Seeds grow linearly and can move by sudden jumping up to a certain size. Further seed growth increases the rate of recontamination which can also be seen in the area coverage over time (**d**). Different scan rates result in different recontamination patterns as shown in x-t slices (**e**). Scanning slowly over a cleaned area results in an even distribution of the recontamination. Conversely, fast scanning produces a stronger increase towards the edges of the pattern. **f** 3D representation of the x, y, t data cubes to illustrate the contamination procedure. **g** Schematic representation of the recontamination mechanisms. While gas-phase diffusion occurs ($M_G$), the cross-linking of molecules ($M_1 + M_2$) moving by surface diffusion is the main process for contamination growth. Once cross-linked molecules become immobile ($M_I$), they can act as a seed for further attachments ($M_N$).

some molecules travel further inside the pattern resulting in a small increase also closer to the center. There is also slightly more accumulated contamination at the corners of the illuminated area due to the high surface curvature. With the scan speed and contamination signal, the mean squared displacement (MSD) of the species on the surface can be estimated. The mean distance measured from the sides of the pattern is reached at 117 nm, which corresponds to a diffusion time of 0.7 ms. With the Einstein equation[32] for two-dimensional motion, the collective diffusion coefficient $D$ for molecules on the surface can be estimated by $D = \frac{x^2}{4t}$, with the time $t$ and the mean displacement $x$ resulting in a diffusion coefficient of $4.88 \times 10^{-12}\,\mathrm{m^2\,s^{-1}}$. Compared with literature values for benzene ($1.5 \times 10^{-9}\,\mathrm{m^2\,s^{-1}}$)[33] and n-alkanes ($1.3$–$3.4 \times 10^{-9}\,\mathrm{m^2\,s^{-1}}$)[34] on graphite, it seems that larger, less mobile specimen must be the prevalent species on the surface. The occurrence of in situ recontamination shows, that even apparently pristine surfaces are never entirely clean in high vacuum. Instead, a diluted film of mobile molecules which cannot be imaged directly due to their speed is always present and has to be taken into account.

**In situ growth of a nanocrystalline graphene layer.** While the exact constituents of the recontamination are not exactly known (with PMMA fragments being a likely candidate), it was shown that they are originating from the sample surface itself. This can be exploited by purposely depositing a large amount of a specific

molecule onto the specimen. By cleaning an area on such a sample, this molecule can then be supplied to the cleaned area via surface diffusion and finally processed with the electron beam resulting in an in situ synthesis of an organic layer. The general idea is schematically shown in Fig. 5a. In this work, we chose copper(II)-tetraphenylporphyrin as a precursor molecule due to its high degree of aromaticity. The copper was added to increase reactivity, and as a potential marker for the imaging of the molecules. The molecule (in dimethoxyethane, DME) was deposited onto the sample by drop-casting, followed by a rinsing and drying step, leaving in-volatile residuals on the sample surface (G + CuTTP, see Supplementary Fig. 8 for Raman spectra). To test the influence of chemical activation and functionalization on the process, the experiments were also conducted on activated graphene with CuTPP (G + actCuTPP) and covalently bound CuTPP by reductive diazotation[35] (G-CuTTP, see experimental section for details). In all cases, a similar behavior to what is described below was observed.

After in situ cleaning, the cleaned area was observed with HRTEM. Starting from the edges of the illuminated area, the growth of a second layer of graphene is observed (Fig. 5b–d; see also Supplementary Movies 5–7). While the first layer only has a single crystallographic orientation, the second layer grows in smaller crystals rotated with respect to the first layer, as inferred from the Moiré pattern. After just under 6 min, a closed second layer is formed, with a third layer starting to grow (an example for a third layer is shown in Supplementary Fig. 9). The

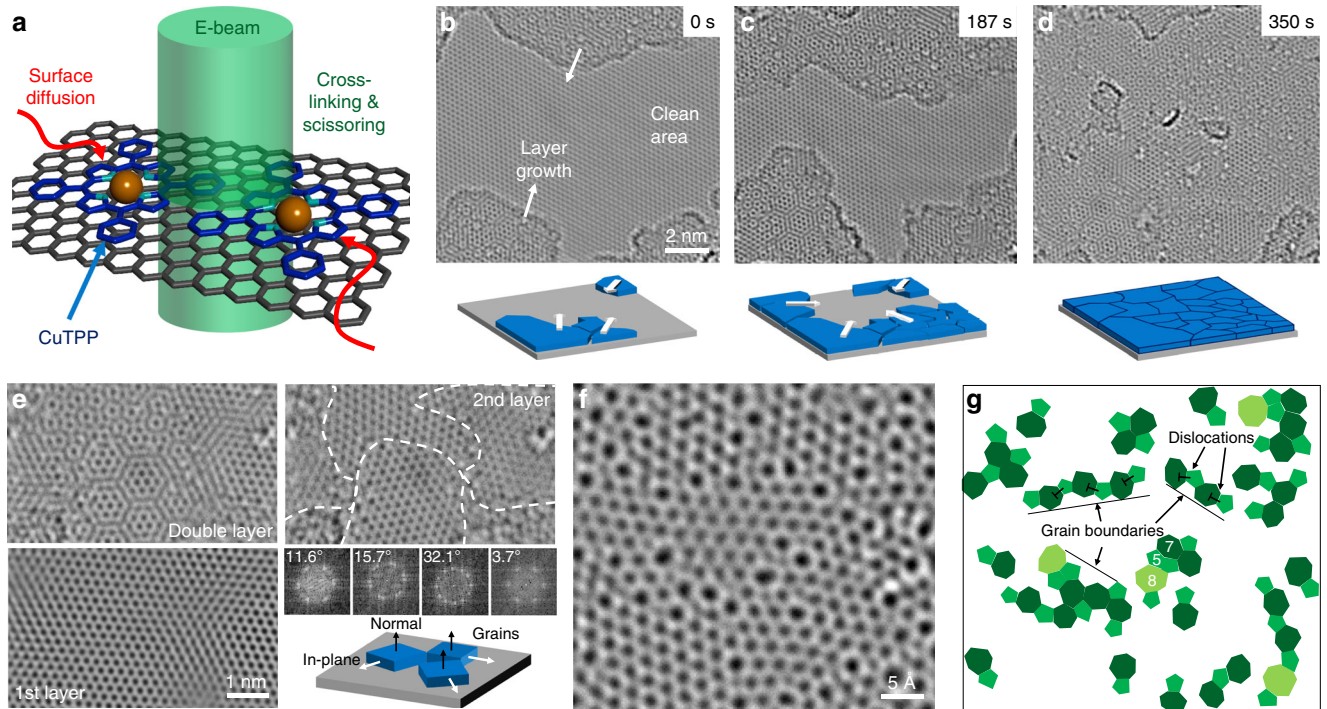

**Fig. 5 Growth of a nanocrystalline graphene layer by supplying a precursor molecule. a** A graphene specimen is covered in CuTPP by drop-casting it onto the surface. After in situ cleaning, the molecules diffuse over the cleaned area. The electron-beam reactions can be induced, resulting in the growth of a second graphene layer (**b**–**d**). See Supplementary Movies 5 for the whole process. The second layer is turbostratically disordered resulting in a pronounced Moiré pattern. By Fourier filtering, this pattern can be disentangled (**e**) revealing small grains in the second layer separated by grain boundaries. The FFT reveals an out-of-plane order with an in-plane disorder. The grain boundaries in the second layer (**f**) have the expected structure of graphene consisting of dislocation walls (**g**).

crystallographic structure of the second layer is analyzed by Fourier filtering, which can computationally separate the two layers as shown in Fig. 5e. The double layer shows a pronounced Moiré pattern, while the individual layers show just one graphene lattice. The second layer consists of small crystalline patches oriented in different ways separated by grain boundaries. The relative rotation can be measured from the FFT showing arbitrary rotation angles. The second layer grows with a [0001] out-of-plane texture and arbitrary in-plane orientation. The grain size is very small, not exceeding a few nanometers. Looking closely at the grain boundary structure of the second layer (Fig. 5f, g), the expected atomic configuration is found with closely spaced dislocations[36]. Besides the normal 5,7 configuration of dislocations[37], also 5,5,8 configurations were found which is a combination of two single dislocations resulting in a larger [1-100] Burgers vector. During the formation of the second layer, even larger rings, such as nine rings with three neighboring five rings are found temporarily. For unrotated bilayer graphene patches, in-plane dislocation contrast and AA-stacking was also observed (see Supplementary Fig. 9). Although it is not clear how the reaction takes place in detail, we assume that the CuTPP is split by radiolysis and/or knock-on induced by the electron beam. The copper, which may have an additional catalytic effect on the growth of the second layer, was not clearly observed during the experiment. The fragments of the TPP are then built into a graphene lattice with hydrogen likely being lost in the process. It is unclear if the nitrogen is incorporated into the lattice at this point. However, it is clearly demonstrated that the targeted supply of a monomolecular species enables the controlled growth of a second nanocrystalline graphene layer. The herein described process is similar to what has been described for the graphitization of amorphous carbon on graphene using electrons[38] or heat[39].

## Discussion

Cleaning of suspended graphene membranes is facilitated by mechanical sweeping using piezo-controlled tips. The membranes are atomically clean after the process in the sense that no stationary adsorbents are visible. However, by analyzing the inner workings of in situ recontamination, a prevalence of mobile molecules even in the cleaned areas is found. This questions what clean actually means in the context of a high-vacuum environment. Molecules may be present on a specimen even though the specimen appears entirely pristine. This is due to the rapid diffusive movement of molecules even at room temperature, which cannot be resolved by electron microscopy so far. On the other hand, this very fact enables the targeted supply of molecules to a specimen area via surface diffusion which can be used for a chemical synthesis as it was demonstrated here: CuTPP was used as a precursor for the growth of a nanocrystalline graphene layer directly observed using HRTEM. We believe that mechanical cleaning will be of significant importance in the further exploration of 2D and nano materials and will be of interest in the broader context of high-vacuum environments.

## Methods

**Suspended graphene sample preparation**. CVD-grown graphene (ACS Materials, "Trivial Transfer Graphene") was transferred to TEM grids as described in the following: The material was immersed in deionized water, releasing it from the substrate. The graphene, still covered in PMMA is then picked up with a filter paper and cut into appropriate pieces (2 × 2 mm²). The material is re-immersed in deionized water and picked up with Quantifoil TEM support grids. Finally, the PMMA is removed by immersing the samples in acetone for 20 s and holding it in acetone vapor atmosphere for 2 h.

**Porphyrin synthesis**. *meso-5,10,15,20-tetraphenylporphyrin-Cu(II) (CuTPP)* and *5-(4-aminophenyl)-10,15,20-(triphenyl) porphyrin (TPP-NH₂)* were synthesized according to literature procedures[40,41].

*5-(4-aminophenyl)-10,15,20-triphenyl-porphyrin Cu(II) (CuTPP-NH₂)*: To a solution of 126 mg (0.2 mmol), 5-(4-aminophenyl)-10,15,20-(triphenyl)porphyrin (TPP-NH₂) in DCM/methanol (4:1, 50 mL) was added 82.7 mg (0.45 mmol) copper(II)-acetate. The mixture was refluxed for 6 h, then diluted with DCM (25 mL), washed with water (75 mL), and 10 % (wt%) solution of aqueous sodium bicarbonate. The organic phase was dried over $Mg_2SO_4$, filtered, and the solvent was evaporated in vacuum and the product was obtained in quantitative yield. $^1$H-NMR (400 MHz, $CDCl_3$, rt): δ (ppm) = 5.28 (s, 2H, $NH_2$), 6.40–6.90 (m, 10H, pyr-H, PhH), 7.30–7.56 (m, 8H, PhH), 7.30–7.80 (m, 9H, PhH). MS (MALDI-TOF, $CHCl_3$, no matrix): $m/z$ 690 [M]$^+$.

*4-(10,15,20-triphenylporphyrin-5-yl-Cu(II))benzenediazonium tetrafluoroborate (CuTPP-N₂⁺ BF₄⁻)*: 0.1 mL (1.59 mmol, 9.94 eq.) HBF₄ (in 48 wt% $H_2O$) and 10 mL acetic acid were added to 109 mg (0.16 mmol, 1 eq.) 5-(4-aminophenyl)-10,15,20-triphenyl-porphyrin Cu(II) (CuTPP-NH₂). Afterward, 0.1 mL (0.75 mmol, 4.69 eq.) isoamylnitrite also dissolved in 5 mL of acetic acid was dropped slowly to the solution. The reaction mixture was quenched with 10 mL diethyl ether after 10 min and stored at −22 °C for 6 h. The solid was filtered off over a 0.2 μm pore filter and washed with cold diethyl ether. The product was yielded in 70% (87.5 mg, 0.11 mmol). MS (MALDI-TOF, $CHCl_3$, no matrix): $m/z$ 675 [M·-N₂]$^+$. IR (ATR, diamond): $\tilde{\nu}$ [cm$^{-1}$] = 3105, 3067, 2359, 2334, 2282, 1578, 1522, 1346, 1076, 1005, 800, 758, 700.

*Preparation of the non-covalent graphene-porphyrin samples (G + actCuTPP and G + CuTPP)*: 150 mg Na/K alloy was dissolved in 5 mL dry and degassed DME and stirred for 1 h in an argon-filled glove box. In order to negatively charge the graphene, 30 μL of the deep blue Na/K solution was dropped on the sample. Afterward, 30 μL *meso-5,10,15,20-tetraphenylporphyrin-Cu(II) (CuTPP)* (6.76 mg, 0.01 mmol dissolved in 1 mL dry and degassed DME) solution was added. The reaction was aborted after 15 min by rinsing off the reactants with 50 μL dry and degassed DME (G + actCuTPP). For the G + CuTPP sample, only 30 μL *meso-5,10,15,20-tetraphenylporphyrin-Cu(II) (CuTPP)* (6.76 mg, 0.01 mmol dissolved in 1 mL DME) solution was dropped onto the graphene and rinsed off after 15 min.

*Preparation of the graphene-porphyrin sample (G-CuTPP)*: 150 mg Na/K alloy was dissolved in 5 mL dry and degassed DME and stirred for 1 h in an argon-filled glove box. In order to negatively charge the graphene, 30 μL of the deep blue Na/K solution was dropped on the sample. Afterward, 30 μL 4-(10,15,20-triphenylporphyrin-5-yl-Cu(II))benzenediazonium tetrafluoroborate (7.90 mg, 0.01 mmol dissolved in 1 mL dry and degassed DME) solution was added. The reaction was aborted after 15 min by rinsing off the reactants with 50 μL dry and degassed DME.

**Scanning electron microscopy**. A FEI Helios NanoLab 660 instrument dual-beam FIB/SEM was used for SEM/STEM imaging and in situ cleaning. The cleaning was performed with 2 MM3A micromanipulators from Kleindiek equipped with commercial W tips (tip diameter of around 100 nm). The microscope was operated at 2 kV for surface-sensitive SE imaging and 20–30 kV for STEM imaging.

**Force measurement**. Force measurements were performed using a Kleindiek STMFA spring table system. The factory calibration of 11 N/m was assumed for the used spring. Spring displacements were tracked with a custom template-matching procedure implemented in python on the basis of openCV.

**Transmission electron microscopy**. For TEM, an aberration-corrected FEI Titan Themis³ 300 electron microscope was used. The cleaning was performed using a Nanofactory STM sample holder outfitted with custom made gold double tips (see Fig. 1). To avoid knock-on damage of graphene, the microscope was set to an operating voltage of 80 kV for both CTEM and HRTEM. In HRTEM, the monochromator was excited to reduce the effect of chromatic aberration. The image series were recorded with a FEI CETA camera. Electron Energy Loss Spectroscopy was performed with a Gatan GIF Quantum at a camera length of 115 mm with a 2.5-mm entrance aperture.

## Data availability
Microscopic and spectroscopic raw data used in this study is available from the corresponding author upon reasonable request.

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

## Acknowledgements
The authors gratefully acknowledge funding by the German Research Foundation (DFG) through the Collaborative Research Center SFB 953 "Synthetic carbon allo-tropes" and the Research Training Group GRK 1896 "In situ microscopy with elec-trons, x-rays and scanning probes." The work has been further supported by the European Union (ERC-2018-StG 804110-2D-PnictoChem to G.A.). G.A. acknowl-edges support by the Generalitat Valenciana (CIDEGENT/2018/001), the DFG (FLAG-ERA AB694/2-1) and the Spanish MINECO (Structures of Excellence Mariá de Maeztu MDM-2015-0538). The Authors would like to thank Mingjian Wu for help with the EELS measurements.

## Author contributions
P.S., C.D., and E.S. wrote the paper. The electron microscopy was performed by C.D. and P.S. D.D. performed the chemical synthesis and the Raman spectroscopic measurements. G.A., F.H., A.H., and E.S. supervised the project.

## Competing interests
The authors declare no competing interests.
