## [Peer Review File · Nature Communications]

Reviewers' comments:

Reviewer #1 (Remarks to the Author):

The authors have investigated the cleaning of 2D materials, primarily suspended CVD-grown graphene, using a nanoscale tip that is used to mechanically remove contamination whilst the material is simultaneously investigated using electron microscopy (EM). This manuscript is well-written and both novel and interesting to the 2D material community by revealing a way to clean these materials for EM investigation, as well as the electron microscopy community which will benefit from the understanding of the contamination process in EM systems. However, there are improvements that could be made to the manuscript.

Firstly, the title is particularly lacking in detail, the fact that this work focusses on 2D materials specifically and is only investigated using electron microscopy systems is neglected and needs to be included in the title.

Secondly, there is a need for more detail about the manipulator itself. For instance, Supp. Info. Figure 1 is supposed to contain 'details on the force measurement', but offers little, not even fully describing the figure itself, which took time to fully understand in terms of measuring the lateral force. The process of how the tip was approached to the graphene layer is not discussed, neither is the calibration of the measurement of the force. Furthermore, it is not understood if the 76.6nN measured is due to the displacement in the sample that occurs as a force is applied with no movement of the manipulator relative to the sample and then there is movement of the manipulator, as the force is large enough to remove the contamination.

Additionally, there is not any chemical characterisation of the contamination in this work, even though the authors have shown that the sample could be removed between instruments. The further work required to conclusively describe the chemical nature of the contamination should be described.

More minor comments include:

Whether the tungsten tip is heated in UHV before use should be included, as this will dictate what the material actually is that is interacting with the graphene/contamination.

There are two instances of where Supp. Info. Figure 3 is referenced on page 7 but this should actually be Supp. Info. Figure 4.

The word 'Moire' should be capitalised.

Reviewer #2 (Remarks to the Author):

In this work by Schweizer et al, a site-specific mechanical cleaning technique upon graphene is carefully demonstrated. By using the in-situ electron microscopy, the cleaning process with the removal of contamination on graphene could be observed, which powerfully proves the efficiency of this method. Also based on this, the authors investigated the process of the recontamination and discuss the synthesis of the nanocrystalline graphene in TEM. Indeed, contamination on graphene are undoubtedly very important issue in the fundamental studies and applications of graphene membranes, and the preparation of clean graphene is definitely of broad interest. Unfortunately, more evidence should be provided before making the clear decision. Importantly, the authors fail to answer the questions in the title "How clean is clean?", and the last two groups of figures seems a little detached from the main idea of the manuscript. Thus, the author should fully address the following issues before the further consideration of this manuscript:

1. As stated by the author, the force between tips and contamination is a key factor for the

cleaning process. Thus, the authors should demonstrate how to achieve precise control of force both in direction and value. And how the value of the force (such as pressure) will affect the efficiency of graphene cleanliness and integrity of graphene membrane should be carefully investigated and discussed.

2. Relying on the in-situ TEM technique, the authors should add more details of the contamination in terms of structures and composition. For instance, atomic structure and origins of the contaminants and secondary contaminants from front to back. As reported previously, there are three origins of contaminants (The airborne contamination reported by Li, Z. et al. *Nat. Mater.* 12, 925-931 (2013); the polymer residues widely studied such as Lin, Y. C. et al. *Nano Lett.* 12, 414-419 (2012), Zhang, Z. et al. *Nat. Commun.* 8, 14560 (2017), Leong, W. S. et al. *Nat Commun.* 10, 867 (2019); and the CVD induced contamination reported recently by Lin, L. et al. *Nat. Commun.* 10, 1912 (2019)), which perhaps with different structures.

3. The authors mentioned that the recontamination seeds prefer to be formed at defects. As we all know, the electron beam would induce new defects on as-cleaned graphene, which in turn will promote the recontamination. Thus, if this is true, the author had better exclude such influence of e-beam when investigating the recontamination process. Thus, the conclusion of spatial preference of recontamination seeds and that of the universality of recontamination would be more trustful.

4. It seems that the recontamination could be removed by this method, but how many times could this process be repeated efficiently? Were the preferred deposition sites and distribution of contaminants kept unchanged? Would the success rate and cleanliness decrease as the increasing the cleaning cycles for the same sample?

5. The authors demonstrated that "the contamination is either pushed to the sides of suspended membrane or it sticks to the cleaning tip...", which is confusing. More detailed evidence about the movement of the contamination during the cleaning should be provided. Considering that the accumulated contamination attached to the tips would affect the following cleaning process, how many times can the tips be used to efficiently clean the graphene? What is the maximum size of the suspended graphene could be cleaned? Are there methods to clean the tips (renew the tip) for new cleaning process?.

6. The authors mentioned the "aged or cross-linked layer" is harder to remove. Do the authors mean that the interaction between contaminants and graphene surface will be stronger as time goes on? The word "cross-linked" is confusing. It seems that the contaminants shown in Figure 2a-c are "cross-linked" network, which, however, are easily removed by the authors. A series quantitative data on "aged" and "cross-linked" would be more meaningful for the further study and controlling of contamination on graphene surface. Besides, why do the contaminants distribute like a network on graphene surface?

7. In a relative reported work, the contamination on bilayer graphene is harder to remove by mechanical method (Algara-Siller, G. et al. *Appl. Phys. Lett.* 104, 153115 (2014)). Would similar conclusions be drawn using this site-specific mechanical method?

8. The authors demonstrated that graphene suffered from no mechanical damage after their mechanical cleaning method proved by HRTEM images. However, the detection range in HRTEM is very limited and I suggest Raman mapping of suspended graphene membrane grid before and after treatment in the same region be conducted for a safe conclusion, because the crystal quality, doping and strain level could be extract from the D-band intensity, G-band position, 2D-band position and full width at half maximum of 2D band.

9. Citation of the previously reported works is not comprehensive. For example, the origin of airborne contamination is not a new discovery, which was reported by Li, Z. et al. 2013 (*Nat. Mater.* 12, 925-931 (2013)). The transformation from contamination to nanocrystalline graphene in TEM was also studied by Börrnert, F. et al. (*Adv. Mater.* 24, 5630-5635 (2012)) and Westenfelder, B. et al. (*Nano Lett.* 11, 5123-5127 (2011)).

Reviewer #3 (Remarks to the Author):

The authors proposed an interesting method of site-specific cleaning of suspended graphene membranes by using a metal tip to remove accumulated contamination, akin to a micro-sized

"broom". However, in my opinion, the potential impact of this work may be rather limited due to several factors that preclude me to recommend the publication of this manuscript in its current form in Nature Communications.

1. Considering that several publications have reported the synthesis of large-area, ultra-clean graphene (for e.g. Lin et al, Towards super-clean graphene, Nat Comm 2019; Leong et al, Paraffin-enabled graphene transfer, Nat Comm 2019; Kim et al, Ultraclean Patterned Transfer of Single-Layer Graphene by Recyclable Pressure Sensitive Adhesive Films, Nano Letters 2015, just to name a few), the authors have not demonstrated that their method is in any way superior, or even equivalent, to the state-of-art, especially given the fact that almost 20% of the membranes they tested had ruptured. In other words, if adsorbents were already minimised in the first place by producing large-area, ultra-clean graphene, wouldn't this imply that their method is somewhat redundant?

2. Just as critically, the authors did not characterise and compare any mechanical or electrical properties of the graphene before and after cleaning. Instead, they focused on recontamination of the suspended graphene and growing new graphene layers on top of it. This does not seem to have any clear links to their proposition of a new way to "clean" graphene, causing the central message of their story to be disjointed and confusing, if there is one. This confusion is made worse by the ambiguous title of "How clean is clean?" because the authors provided no clear answer either. How clean is clean to achieve what exactly?

3. Similarly, for their second proposition of growing new graphene layers on top of the first, it is unclear how this is beneficial/superior to any of the current ultra-clean techniques for producing graphene. The novelty here is rather questionable.

Therefore, while I do not dispute the technical accuracy and feasibility of the techniques proposed by the authors which are well-written and detailed, there are no clear benefits to applying their methodologies and hence I do not find that the manuscript is suitable for publication in Nature Communications.

Reviewer 1

We would like to thank reviewer 1 for her/his generous support of our manuscript which she/he called “[...]well-written, novel and interesting[.]”. We gladly address the issues raised point by point below.

- 1. Firstly, the title is particularly lacking in detail, the fact that this work focusses on 2D materials specifically and is only investigated using electron microscopy systems is neglected and needs to be included in the title.**

As a response to this valid point, we decided to change the title of the manuscript to include more details of our experimental work. The title now reads “Mechanical cleaning of graphene using *in situ* electron microscopy”.

- 2. “Secondly, there is a need for more detail about the manipulator itself. For instance, Supp. Info. Figure 1 is supposed to contain ‘details on the force measurement’, but offers little, not even fully describing the figure itself, which took time to fully understand in terms of measuring the lateral force. The process of how the tip was approached to the graphene layer is not discussed, neither is the calibration of the measurement of the force. [...]”**

We thank the reviewer for this comment and her/his interest in the details of our force measurement. As a response (also to reviewer 2 point 1), we decided to expand upon the description of the force measurement in both the main manuscript, the supporting information and the methods section to clarify the questions raised. In addition, we performed further experiments to also measure the normal force during cleaning. These measurements are mentioned in Figure 2 and expanded upon in supporting Figure 2 (with an additional supporting

movie 1 b). We also included a further supporting figure (SI 3), in which we measure the yield strength of the graphene membranes as a reference for the cleaning force. We are convinced that these additions further improve our manuscript.

To answer the concrete question of the reviewer: **“Furthermore, it is not understood if the 76.6nN measured is due to the displacement in the sample that occurs as a force is applied with no movement of the manipulator relative to the sample and then there is movement of the manipulator, as the force is large enough to remove the contamination.”**:

The force is measured using template matching on an un-deformed feature on the sample (now demonstrated in supporting Figure 2), which is mounted on a spring table. When a force is acting on the specimen, the spring table deflects accordingly, which results in a small shift of the feature used for tracking. In the lateral direction sample deformations can be neglected as the movement is mainly governed by the spring table deflection. In the normal force direction, the support film deforms significantly (see supporting Figure 3). In this case, the force signal can only be measured by tracking the deflection of the rigid TEM grid itself. The manipulator tip, which applies the load, is also tracked during the measurement. The tip only moves if the force is large enough to remove the contamination, which results in the around 76 nN average cleaning force. This force is dependent on the area that is cleaned at the same time, as a larger area would also require a larger force.

- 3. Additionally, there is not any chemical characterisation of the contamination in this work, even though the authors have shown that the sample could be removed between instruments. The further work required to conclusively describe the chemical nature of the contamination should be described.**

As a response to this point raised by the reviewer (and also in response to reviewer 2 point 2), we performed electron energy loss spectroscopy measurements of as-prepared and cleaned membranes. A spectrum has been included in Figure 3 for reference and described in the text with added references. The clear oxygen signal in the as-prepared state strongly hints towards the presence of PMMA residuals, which was the polymer used for transfer onto the TEM grids. After cleaning and exposing the sample to ambient conditions, the oxygen signal cannot be detected anymore, which indicates that the airborne contaminants are of a different chemical variety. In addition to the disappearance of the oxygen signal, also a plasmon peak shift to lower energy values is noticeable after cleaning. The lower value corresponds to graphene while the higher value is more indicative of the PMMA residuals. We hope that the inclusion of spectroscopy adequately addresses the question regarding the chemical nature of contamination.

More minor comments include:

- 4. Whether the tungsten tip is heated in UHV before use should be included, as this will dictate what the material actually is that is interacting with the graphene/contamination.**

The tips were not subjected to further treatment. We included the information at an appropriate point in the manuscript.

5. **There are two instances of where Supp. Info. Figure 3 is referenced on page 7 but this should actually be Supp. Info. Figure 4.**

Thanks for noticing this oversight on our behalf. We changed the reference.

6. **The word 'Moire' should be capitalised.**

We changed all instances of Moiré to be capitalized.

Reviewer 2

We would like to thank reviewer 2 for her/his general support of our manuscript. Her/his comments are addressed point-by-point below.

1. **As stated by the author, the force between tips and contamination is a key factor for the cleaning process. Thus, the authors should demonstrate how to achieve precise control of force both in direction and value. And how the value of the force (such as pressure) will affect the efficiency of graphene cleanness and integrity of graphene membrane should be carefully investigated and discussed.**

We thank the reviewer for raising a similar point as reviewer 1 (point 2). As mentioned in the answer to reviewer 1, we expanded upon the force measurements by also including values for the normal force during cleaning and comparing it against the strength of graphene. These values are discussed at appropriate points in the manuscript and further elaborated on in SI 2 & 3. We further included a general pressure requirement for cleaning that relates the measured forces to the cleaned area. We hope these additions address the point raised by the reviewer.

2. **Relying on the in-situ TEM technique, the authors should add more details of the contamination in terms of structures and composition. For instance, atomic structure and origins of the contaminants and secondary contaminants from front to back. As reported previously, there are three origins of contaminants (The airborne contamination reported by Li, Z. et al. Nat. Mater. 12, 925-931 (2013); the polymer residues widely studied such as Lin, Y. C. et al. Nano Lett. 12, 414-419 (2012), Zhang, Z. et al. Nat. Commun. 8, 14560 (2017), Leong, W. S. et al. Nat Commun. 10, 867 (2019); and the CVD induced contamination reported recently by Lin, L. et al. Nat. Commun. 10, 1912 (2019)), which perhaps with different structures.**

To address this point we performed additional EELS measurements of as-prepared and mechanically cleaned membranes (results are now included in Figure 2 and described in the text with added references). The oxygen signal in the as-prepared membranes points towards the majority of residuals originating from PMMA, which is expected since that polymer was used for the transfer to the TEM grids and is widely recognized as the major contaminant source when used (in accordance with the references the reviewer mentions). After cleaning and exposing the membranes to ambient conditions, the oxygen signal vanishes. This means that the airborne contamination contains little to no oxygen and is comprised mostly of hydrocarbons (in agreement with Li, Z. et al. Nat. Mater (2013)). We think that the recently proposed CVD induced

contamination does not play a significant role in our study, as we see clear signs of PMMA being the dominant source of contamination for as-prepared membranes.

- 3. The authors mentioned that the recontamination seeds prefer to be formed at defects. As we all know, the electron beam would induce new defects on as-cleaned graphene, which in turn will promote the recontamination. Thus, if this is true, the author had better exclude such influence of e-beam when investigating the recontamination process. Thus, the conclusion of spatial preference of recontamination seeds and that of the universality of recontamination would be more trustful.**

The comment about the recontamination being preferred at defects mainly refers to the observation around basal dislocations (see supporting figure 7). These defects are intrinsic to bilayer graphene and can in principle not be created by electron irradiation. Edges of bilayer regions transitioning to monolayer region also show an increase in contamination rate. Point defects have not been explicitly studied as starting points for contamination seeds.

We do not entirely share the reviewers opinion that **“As we all know, the electron beam would induce new defects on as-cleaned graphene”**. It has been conclusively shown that under the right conditions Graphene can be extremely stable in electron microscopy (e.g. J. Meyer et al. PRL (2012)). In this study we always stayed at energies of 80 keV (TEM), 20 keV (SEM) and in some cases even 2 keV, all of which are well below the knock-on threshold. During the HRTEM observation of cleaned areas we did not observe the formation of additional defects. We therefore confidently rule out the electron beam induced formation of defects as a significant influence to the recontamination.

- 4. It seems that the recontamination could be removed by this method, but how many times could this process be repeated efficiently? Were the preferred deposition sites and distribution of contaminants kept unchanged? Would the success rate and cleanness decrease as the increasing the cleaning cycles for the same sample?**

We like to thank the reviewer for encouraging us to provide more details on repeated cleaning of recontamination. In principle there is no limit for how many times a membrane can be cleaned. In practical terms, cleaning of one membrane more than a few times might have diminishing returns because the process can be admittedly time consuming. The spatial distribution of recontamination in the absence of defects is essentially random and does not depend on the number of cleaning steps. The success rate for repeated cleaning does not decrease compared to the initial cleaning step. If a membrane has already been cleaned once, it is likely that the process can be repeated since it has proven to be mechanically stable enough. We added supporting figure 4 to highlight the removal of a purposely-deposited carbon piece after cleaning, also in response to point 5.

- 5. The authors demonstrated that “the contamination is either pushed to the sides of suspended membrane or it sticks to the cleaning tip...”, which is confusing. More detailed evidence about the movement of the contamination during the cleaning should be provided.**

To address this point we gladly clarified this part in the manuscript. In addition, we included a new supporting figure (4) to highlight the behavior of rigid contamination, which glides over the surface during cleaning as a whole.

Considering that the accumulated contamination attached to the tips would affect the following cleaning process, how many times can the tips be used to efficiently clean the graphene?

Tips can be used as long as the accumulated contamination on them is not in the same size range as the membrane you are trying to clean. In that case, judging the point of contact becomes difficult.

Are there methods to clean the tips (renew the tip) for new cleaning process?

Tips can be cleaned again by purposely ramming them against a substrate, however this process is rather unreliable. Alternatively FIB milling can be used to clean the tips.

What is the maximum size of the suspended graphene could be cleaned? We mostly used holes in quantifoil film with a size of 1, 1.5 and 2 μm^2 which could be cleaned entirely. However, we also successfully tried larger windows (7 by 20 μm) in epitaxially grown graphene (see our earlier work on how to prepare membranes for TEM in this case: Waldmann et al. ACS Nano, 7, 5 (2013)) but did not completely remove the contamination over the whole area.

- 6. The authors mentioned the “aged or cross-linked layer” is harder to remove. Do the authors mean that the interaction between contaminants and graphene surface will be stronger as time goes on? The word “cross-linked” is confusing. It seems that the contaminants shown in Figure 2a-c are “cross-linked” network, which, however, are easily removed by the authors. A series quantitative data on “aged” and “cross-linked” would be more meaningful for the further study and controlling of contamination on graphene surface.**

By “aged or cross-linked layer”, we mean that the layer itself will become stronger due to certain processes. The interaction to the substrate should not change dramatically as the chemical nature of the interaction is not changed. Aging of contamination layers simply refers to an increase in thickness due airborne deposition often observed in old samples. Cross-linking occurs under electron beam irradiation (as shown in the manuscript and generally exploit in negative resist e-beam lithography). Further cross-linking mechanisms could involve visible light when samples are stored near windows. Supporting figure 4 shows the case of a thick and cohesive piece of contamination which can only be removed as a whole. The force per area is however, the same as in the case of small contamination patches.

Besides, why do the contaminants distribute like a network on graphene surface?

That is a keen observation. At this point we unfortunately cannot give a satisfying answer to this question.

- 7. In a relative reported work, the contamination on bilayer graphene is harder to remove by mechanical method (Algara-Siller, G. et al. Appl. Phys. Lett. 104, 153115 (2014).). Would similar conclusions be drawn using this site-specific mechanical method?**

The layer number does not impact the ability to clean the membrane in our case. If anything, the added mechanical strength, makes it easier to clean bi-/multilayer graphene. We added a sentence in the manuscript to clarify this.

- 8. The authors demonstrated that graphene suffered from no mechanical damage after their mechanical cleaning method proved by HRTEM images. However, the detection range in HRTEM is very limited and I suggest Raman mapping of suspended graphene membrane grid before and after treatment in the same region be conducted for a safe conclusion, because the crystal quality, doping and strain level could be extract from the D-band intensity, G-band position, 2D-band position and full width at half maximum of 2D band.**

We thank the reviewer for suggesting to crosscheck potential damage by spectroscopic means. We performed Raman spectroscopy on as-prepared as well as mechanically cleaned membranes and included a spectrum in Figure 2 and additional details in supporting Figure 1. The most striking feature of the spectra after cleaning is the lack of significant D-band intensity. This already demonstrates that no substantial amount of defects were introduced by the cleaning method. We further did not find significant shifts of the G or 2D bands indicating an increase in strain after cleaning or charge transfer effects. We hope this addition in the manuscript adequately addresses the points raised by the reviewer.

- 9. Citation of the previously reported works is not comprehensive. For example, the origin of airborne contamination is not a new discovery, which was reported by Li, Z. et al. 2013 (Nat. Mater. 12, 925-931 (2013)). The transformation from contamination to nanocrystalline graphene in TEM was also studied by Börrnert, F. et al. (Adv. Mater. 24, 5630-5635 (2012)) and Westenfelder, B. et al. (Nano Lett. 11, 5123-5127 (2011)).**

We thank the reviewer for providing additional references for consideration. The references have been included in the discussion of the manuscript at appropriate points.

Reviewer 3

While we thank reviewer 3 for her/his time to read our manuscript and write a review, we are somewhat surprised by his/her conclusion. He/she acknowledges that he/she **“do[es] not dispute the technical accuracy and feasibility of the techniques proposed by the authors which are well-written and detailed”** but then goes on to say that **“there are no clear benefits to applying their methodologies.”** We think that this statement is premature and should be revised in particular when looking at the concrete points raised.

- 1. Considering that several publications have reported the synthesis of large-area, ultra-clean graphene (for e.g. Lin et al, Towards super-clean graphene, Nat Comm 2019; Leong et al, Paraffin-enabled graphene transfer, Nat Comm 2019; Kim et al, Ultraclean Patterned Transfer of Single-Layer Graphene by Recyclable Pressure Sensitive Adhesive Films, Nano Letters 2015, just to name a few), the authors have not demonstrated that their method is in any way superior, or even equivalent, to the state-of-art, especially given the fact that almost 20% of the membranes they tested had ruptured. In other words, if adsorbents were already minimised in the first place by producing large-area, ultra-clean graphene, wouldn't this imply that their method is somewhat redundant?**

We have the impression that the reviewer might have entirely missed the point of our study. None of the papers she/he cited, as being redundant to our work, are actually concerned with the removal of pre-existing contamination. In addition, the cited papers are even somewhat contradictory to each other, which already shows that there is no clear consensus reached about the origins and removal of contamination on graphene. While *Leong et al. (Nat Comm 2019)*, *Kim et al. (Nano Letters 2015)* and *Zhang et al. (Nat Comm 2017)* make out **transfer-induced** contamination to be the main issue, which they try to reduce by means of a different polymer and process, *Lin et al. (Nat Comm 2019)* claim to prove that contamination **originates almost entirely from the CVD growth process**. Interestingly, in the same paper the authors demonstrate contamination originating from the transfer to target substrates, which they are not able to get rid of. Further, at ambient conditions, airborne contaminants will always be present on graphene, no matter how 'clean' it is manufactured or transferred (as described by *Li et al. Nat. Mater. (2013)*). In all of these studies, contamination ends up on the graphene surface at the end. We therefore wonder, how one could come to the conclusion, that being able to entirely get rid of this contamination would be merely redundant. There will always be a need for methods of post-cleaning even for pristine graphene. Additional need for cleaning methods arises when graphene is further processed, for instance via a wet-chemical synthesis route, tested in certain environments or stored for extended periods of time.

2. Just as critically, the authors did not characterise and compare any mechanical or electrical properties of the graphene before and after cleaning. Instead, they focused on recontamination of the suspended graphene and growing new graphene layers on top of it. This does not seem to have any clear links to their proposition of a new way to "clean" graphene, causing the central message of their story to be disjointed and confusing, if there is one. This confusion is made worse by the ambiguous title of "How clean is clean?" because the authors provided no clear answer either. How clean is clean to achieve what exactly?

While we agree that the title might have been an unfortunate choice, which we happily change, we do not agree with the rest of the reviewers sentiment. Mechanical data has in fact been provided in the manuscript and interpreted appropriately. While the contamination does not have an impact on the intrinsic strength of graphene, it can have an influence on the mobility of defects in bilayer graphene as already shown by *Schweizer et al. (Science Advances (2018) SI 3)*. It is already well established that contamination has a detrimental effect on the electrical properties of graphene. While we think that electrical measurements could be explored in a further study, they are beyond the scope of the present work and would not benefit the current manuscript.

3. Similarly, for their second proposition of growing new graphene layers on top of the first, it is unclear how this is beneficial/superior to any of the current ultra-clean techniques for producing graphene. The novelty here is rather questionable.

The reviewer seems to have missed the part of our study dealing with recontamination and its origin especially in electron microscopy and high vacuum environments. The final point in our study about the growth of an additional graphene layer firstly confirms the previously stated prevalence of surface diffusion on graphene and TEM samples in general. It is secondly a way of producing nanocrystalline graphene on top of large-crystalline graphene, a feat that currently cannot be done with conventional techniques. Finally, it may be a pathway towards the implementation of heteroatoms and defects into a graphene lattice. We therefore reject the opinion of the reviewer that there is no benefit of our technique.

Reviewers' comments:

Reviewer #1 (Remarks to the Author):

I thank the authors for their thorough response to my comments and the extra work and measurements they have put in to improving the manuscript.

One further comment I have on the changes is that the 'cleaned' Raman spectra in Figure 2i appears to show a shoulder on the 2D peak. This could be due to some kind of peak splitting, indicating a change in the material, but I suspect it may be the way the spectra has been plotted. If this is the case, I would recommend replotting this spectra so this apparent shoulder isn't observed as it would appear to contradict the conclusions that the two Raman spectra are similar, other than the D peak change.

Also, I would make sure that the authors reread the changes to check for grammar mistakes and typos, as I spotted some, including two references to Figure 2g on page 7 that should actually refer to Figure 3g.

Reviewer #2 (Remarks to the Author):

In this updated paper, Schweizer et al have revised the title, some content in the main text and supporting information. A clearer story about mechanical cleaning of graphene surface is provided. However, I think this work is still premature to be published in Nature Communications, the high-impact top journal. A few comments are listed below:

1. Generally speaking, title gives information about originality and achievement of a work. However, the in-situ mechanical cleaning method demonstrated in this manuscript does not represent an important progress in this field. On the one hand, the mechanical cleaning of graphene surface has been reported by several groups, e.g. Mechanical Cleaning of Graphene, *Appl. Phys. Lett.* 2012, 100, 073110; Cleaning Graphene Using Atomic Force Microscope, *J. Appl. Phys.* 2012, 111, 064904; A Force-Engineered Lint Roller for Superclean Graphene, *Adv. Mater.* 2019, 31, 1902978 and so on. The quantitative analysis of the force is worthy to be affirmed. But no new principles are put forwards. On the other hand, in-situ technique in SEM and TEM chamber is not a big challenge for EM experts, as can be concluded from previous literatures, e.g. An Electromechanical Material Testing System for in Situ Electron Microscopy and Applications, *PNAS*, 2005, 102, 14507; Direct Observation of Weakened Interface Clamping Effect Enabled Ferroelastic Domain Switching, *Acta Mater.* 2019, 171,184e189.

2. The author neglected our suggestion in adjusting the logic of this manuscript. In the first round of the review, we commented that the last two figures did not have close connection with the first two figures. Necessary relevance in all the figures in the main text should be clearly displayed. However, no changes have been conducted in the revised manuscript. What's worse, separation of the original Figure 2 into updated Figures 2 and 3 is not so necessary, especially with the figure topics being written as "In situ mechanical cleaning approach in the TEM/SEM".

3. The Raman data added in Figure 2i and Figure S1e cannot adequately support the authors' conclusion. Firstly, the bad signal-to-noise ratio can be experimentally avoided, for example, by increasing integration time, using near-field Raman spectrograph or using suspended graphene with larger size to reducing the impact from the underlying substrates. Secondly, the origin of D peak in the reference spectrum lacks explanation. Does the contamination contribute to the appearance of defective peak? If so, would PMMA residues or airborne contamination has the same effect? Actually, D peak is not very common in the reported works, even though graphene samples are unclean. Thirdly, the authors failed to supply Raman mapping data of D peaks and did not give any explanation, which is not so eligible. To some extent, it shows the attitude of the authors in their response to the reviewers.

4. The scaling up potential of this technique is not satisfactory enough. In response to our 5th question, they admitted that using this methodology, contamination on graphene films in larger windows (7 by 20 μm) cannot be cleaned completely. Thus, it seems that the potential application fields of this method are seriously limited.

Reviewer #3 (Remarks to the Author):

After considering the change in title and the authors' careful responses to the reviewers, I find that the train of thought is made much clearer for the broader audience to follow and I am happy to recommend its publication. However, I would also recommend including the following key points that the authors raised so that the article's impact is clearly stated:

1. "There will always be a need for methods of post-cleaning even for pristine graphene. Additional need for cleaning methods arises when graphene is further processed, for instance via a wet-chemical synthesis route, tested in certain environments or stored for extended periods of time."

2. "...firstly confirms the previously stated prevalence of surface diffusion on graphene and TEM samples in general. It is secondly a way of producing nanocrystalline graphene on top of large-crystalline graphene, a feat that currently cannot be done with conventional techniques. Finally, it may be a pathway towards the implementation of heteroatoms and defects into a graphene lattice."

Minor Amendments:

At line 208 on page 9, there seems to be an error in the statement: "A preferential formation of seeds at defects, such as dislocations graphene is observed".